# Comparison of Mechanical Properties of Composites Reinforced with Technical Embroidery, UD and Woven Fabric Made of Flax Fibers

**DOI:** 10.3390/ma15217469

**Published:** 2022-10-25

**Authors:** Agata Poniecka, Marcin Barburski, David Ranz, Jesús Cuartero, Ramon Miralbes

**Affiliations:** 1Institute of Architecture of Textiles, Faculty of Material Technologies and Textile Design, Lodz University of Technology, Zeromskiego 116, 90-543 Lodz, Poland; 2Department of Design and Manufacture Engineering, University of Zaragoza, C/María de Luna, 50018 Zaragoza, Spain; 3Department of Mechanical Engineering, University of Zaragoza, C/María de Luna, 50018 Zaragoza, Spain

**Keywords:** composites, technical embroidery, natural fibres, flax, directional orientation, automated lay-up

## Abstract

The main purpose of the article is to present the possibilities of producing composite reinforcement with the use of a computer embroidery machine. The study below presents the results of strength tests of composites containing technical embroidery, woven fabric, and UD fabric as the reinforcement. Each of the samples was made of the same material—flax roving. The samples differed from each other in the arrangement of layers in the reinforcement. The composites were made using the infusion method with epoxy resin. The embroidery was made on a ZSK embroidery machine, type JCZA 0109-550. A total of 12 types of composites were produced and tested. The test material was subjected to strength tests—tensile strength, tensile elongation, and shear strength, on the INSTRON machine. As the research showed, the use of technical embroidery as a composite reinforcement increases its tensile strength. Furthermore, the use of embroidery is a vertical reinforcement of the composite and prevents the formation of interlayer cracks. The technology of technical embroidery allows for optimizing the mechanical values of the composite reinforcement.

## 1. Introduction

The world is currently facing the risk of a climate catastrophe. The continuously growing amount of waste is causing companies to look for new manufacturing solutions to minimize their production waste. World economies are gradually turning to the idea of sustainable development. A significant aspect of such an economy is the reduce–reuse–recycle principle, i.e., limiting production and waste, using already produced goods as long as possible, and recycling those items which are no longer usable. The use of technical embroidery technology is in line with the idea of sustainable development. Its main purpose is the production of precisely planned shapes and mechanical properties—the waste from patterns is therefore reduced to a minimum. For this reason, the research presented below focuses on the use of technical embroidery as a reinforcement of composites.

Thanks to the full freedom of the medium direction on the substrate offered by the computerized embroidery machine, technical embroidery can be fully customized. Automation of the embroidery process gives full control over the execution of the pattern and the consumption of raw materials. A computerized embroidery machine uses a technique known as Tailored Fiber Placement (TFP). This technology allows placing a medium on the surface of a flat textile product in any direction of the X and Y-axis. By overlapping the embroidery layers, it is also possible to achieve a certain dimension in the Z-axis direction, but this is quite limited by the technical capabilities of the embroidery machine. The height of the layers can reach up to 8 mm. TFP enables the creation of any previously designed pattern, which translates into a very high degree of production waste reduction. In the case of embroidery, the process of cutting out the material is not necessary, as the finished product is received. The only waste is the interlining or other types of backing on which the embroidery was made, and this can also be reduced by using the right size of hoop. Various materials can be used in the embroidery process, e.g., electric wires, optics, glass or carbon fibers, and electrically conductive yarns [1]. TFP technology is mainly used to create heating mats (e.g., in car seats), shielding, conductive interconnections, textile sensors and interfaces.

Moreover, an unquestionable advantage of embroidery is the high dimensional stability of produced preforms (thanks to reinforcing the system with a fixing thread both in the horizontal and vertical plane), as well as excellent repeatability of the production process [2].

The limitations of the embroidery machine include its working speed (maximum 850 punctures/minute), which gives a lower efficiency than the production of woven fabrics. Another limitation is the type of fastening thread—it must be an elastic thread; the linen thread is too stiff and breaks during the embroidery production process.

These days, technical embroidery is mainly used in textronics. Sensors, digital components, or electronics can be attached to flat textile products. Technical embroidery can be successfully used to create antennas. It is also widely used in textronics—an embroidered thread connects from, e.g., a power source to a sensor or another electronic element. Embroidery can also be used as an alternative to solid copper to make a coil for unilateral nuclear magnetic resonance systems [3,4,5,6,7].

In the following research, the use of flax as a raw material for composite reinforcement was dictated by the principles of sustainable development. Compared to other natural fibers, flax is the most ecological in cultivation, while at the same time having the highest strength properties [8]. Additionally, the cultivation of flax is pollution-free, and the waste generated during fiber processing is non-toxic and harmless [9]. Therefore, it is an excellent alternative in the production of composites, compared to the previously used synthetic or glass fibers [10,11,12]. Moreover, the mechanical properties of composites containing natural fibers are comparable to those of composites made of glass fibers [13]. The disadvantage of flax, compared to synthetic fibers, in the composite manufacturing process is its hydrophilicity. However, flax, as the only natural fiber, has increased tensile strength in the wet state compared to the dry state [14].

According to the Nova Institute and FNR, flax usage as a reinforcement material for composites in engineering, furniture, consumer goods, and automotive applications has increased significantly since 2012 [15]. Composites containing natural fibers are lighter, and as a result, a car built with such material consumes less energy while driving, while emitting less pollution [16,17]. Compared to 2012, almost three times more such composites were produced and used in 2020. This shows that the demand for composites reinforced with natural resources is growing [18,19].

The woven fabric consists of two sets of threads perpendicular to each other. For this reason, it is characterized by similar strength values for the force acting both along the weft and the warp. However, its strength values are weaker for forces acting at different angles [20,21,22,23]. Technical embroidery technology, on the other hand, makes it possible to arrange the fibers at precisely planned angles. This makes it possible to optimize the strength properties of the manufactured product [24].

In the literature, there is a time gap in publications on technical embroidery as reinforcement of composites. There can be found publications from the late 1990s and early 2000s. For example, in 1996, a study of samples reinforced with technical embroidery was carried out. The study showed that embroidery strengthens the structure of the components. However, the composites themselves have not been tested [25].

Another study, carried out in 1999, describing the use of embroidery as reinforcement of composites used the technology of flat embroidery—the fabric was reinforced with embroidery. In the case of the following tests, the entire structure was made of three-dimensional embroidery [26]. Similarly, in a study conducted in 2016—embroidery was also used only to reinforce the fabric [27]. There have been no publications in the last few years that fully exhaust this issue. For this reason, it was decided to take up this research topic. It has been proven in the course of the research described below that embroidery can successfully replace other types of reinforcements in the production of composites. Moreover, the use of technical embroidery as a composite reinforcement increases its tensile strength and prevents the formation of interlayer cracks. Moreover, the technology of technical embroidery allows optimizing the mechanical values of the composite reinforcement.

## 2. Materials

The subject of the research is composites containing five types of reinforcements:Technical embroidery, stitch length 2 mm, made of flax roving;Technical embroidery, stitch length 4 mm, made of flax roving;Technical embroidery, stitch length 8 mm, made of flax roving;Unidirectional flax roving arrangement, also made on an embroidery machine (UD fabric);Flax woven fabric.

Each reinforcement system consisted of four layers. In addition, each of the five variants of reinforcements was produced in three directions of roving alignment. The individual sample variants are shown in Table 1. A total of 12 types of reinforcements were produced.

### 2.1. Production of Embroidery and UD Fabric

The samples were made as follows: The design of embroidery and UD fabric was made in the GiS BasePack version 10 program (GiS, Lenningen, Germany). The embroidery and UD fabrics were made on a ZSK embroidery machine, type JCZA 0109-550 (Figure 1a) (ZSK and SWF Embroidery Equipment, Earth City, MO, USA); it is equipped with a W-type head designed for technical embroidery (Figure 1b). This type of embroidery consists in placing a medium on a flat textile product and then attaching it with a zig-zag stitch. This technology is called Tailored Fibre Placement.

Technical embroidery can be classified as an additive manufacturing technology. It is a technique for the production of three-dimensional objects based on their computer models, which consists of interconnecting successive layers of material. Compared to the subtractive method, in which the material is removed by, for example, machining, grinding or drilling, it has many advantages. These are, firstly, less material losses, greater production flexibility, ease of manufacturing objects with complex shapes and high repeatability. On the other hand, the disadvantages of this type of production include technological limitations and lower efficiency compared to the production of composite reinforcements from woven fabrics. In the case of the ZSK type JCZA 0109-550 embroidery machine used, the maximum height of the technical embroidery may be 8 mm.The embroidery and UD material were made of Safilin’s flax roving with a linear mass of 400 tex. It was fastened with Gunold’s polyamide monofilament with a linear mass of 11 tex. For embroidery, the zig-zag stitch lengths were 2 mm, 4 mm, and 8 mm, and the width was 1.2 mm. For the UD material, the same flax roving was attached to the surface only at the ends of the sample, as shown in Figure 2b. A unidirectional alignment in a given layer was performed. This was to imitate the fabric, but without interlacing the threads in a single layer.

The tensile strength of the flax roving was 7.37 cN/tex, and the tensile elongation was 1.69%. Both embroidery and UD fabric were made on a base of cotton fabric with an area weight of 280 g/m^2^ and non-woven fabric with an area weight of 35 g/m^2^.

The flax woven fabric (Figure 3) was made from the same roving used for the embroidery. The surface mass of the fabric was 400 g/m^2^.

### 2.2. Composite Manufacturing

Composites were then made from all the samples using the resin infusion method. This method consisted in laying the previously prepared samples on an aluminum mold, covering them with a polyester peel-ply and a guide mesh, and then closing the system with a vacuum bag and sealing it with tacky tape. Tubes were attached to the two opposite edges of the system to exhaust the air and supply the resin mixture. The resin system consisted of SR GreenPoxy 33 epoxy resin and SD4772 hardener at a ratio of 100:32.

Epoxy resin is one of the most commonly used resins during the composite manufacturing process. It has good mechanical and chemical properties. It has high hardness, high-temperature resistance, and water resistance. Compared to vinylester and polyester resins, epoxy resin has much lower shrinkage and does not emit harmful volatile substances during curing. It is also characterized by excellent adhesion properties [10].

The process of air extraction from the composite and the infusion of the resin mixture was carried out simultaneously until the whole composite was saturated, while the vacuum of 1 Bar was maintained until the composite cured (minimum 4 h). To drain the excess resin mixture from the composite, a resin trap was used. It was placed in the laminate-vacuum pump path. After resin soaking, the entire system was cured for 4 h at 50 °C, as the technical info of resin states.

Volume fractions of prepared composites are shown in Table 2.

### 2.3. Test Samples Preparation

Once the resin had fully cured (after 24 h), the finished composite was removed from the mold and prepared for testing. In the case of samples intended for tensile testing, tabs made of fiberglass were attached to the edges of properly prepared composite. This was to reduce stress concentrators in the grips. All composites were then cut into samples of standard size. The sample preparation for testing is shown in Figure 4.

## 3. Methods

After that, tensile strength, tensile elongation, and short beam bending strength tests were performed on the generated samples.

According to the PN-EN ISO 527-4 standard [28], the tensile strength and tensile elongation tests were performed. The test involved stretching the ready samples at a constant speed until the breakage was attained. The relative elongation at maximum force, maximum force, breaking force and relative elongation at break values were collected during the testing.

The tests were carried out using a 100 kN load cell on an INSTRON universal testing machine, model 8032. The test was conducted at a 1mm/min velocity. A 50 mm gauge length extensometer was used to measure the specimens’ elongation. The test’s parameters are presented in Table 3.

The test findings were presented in the form of numerical data and a graph of tensile stress as a function of elongation.

According to the ASTM D2344 standard, the interlaminar shear strength characteristics were assessed [29]. The test was carried out in accordance with the three-point bending test, with the load registered. This test was conducted until the sample was destroyed or until the support was in direct contact with the lower surface of the specimen. The test determined the maximum load, the short beam strength, and the different types of failure models. Test parameters are presented in Table 4.

Tests were carried out on the same INSTRON universal testing machine, and the composites were produced at The Department of Mechanical Engineering of the University of Zaragoza.

## 4. Discussion

### 4.1. Tensile Strength and Tensile Elongation

The maximum strength of produced composites is presented in Figure 5.

Among the composites with a 0° arrangement of reinforcement, the variant with a stitch length of 4 mm had the highest tensile strength. It was about 10% more durable than the 8 mm variant and 15% more durable than the 2 mm variant. Therefore, it can be observed that too many needle stitches during embroidery (2 mm variant) negatively influence the composite strength. Probably, the puncture of the needle causes damage to the structure of the sample, which reduces its strength. Moreover, a stitch that is too wide (8 mm) does not improve the composite strength. In this case, there is less monofilament that holds the roving, so less material is involved in the stretching process than in the case of samples containing a stitch length of 4 mm, which reduces the strength of the entire sample. The UD and fabric variants showed significantly lower tensile strength. The UD variant had only 60% of the strength of the 4 mm variant, while the fabric had only 46%. For the UD variant, the low durability is due to the chaotic arrangement of the roving in the sample. Although the roving was attached in the longitudinal direction during the embroidery process, the waving and movement of the roving between the layers could not be avoided. The woven fabric, on the other hand, has two arrangements of threads, interlacing each other at right angles along the length of the sample. Theoretically, with four layers of woven fabric, there are eight threads involved in the tension, but half of the threads are at 90° to the tensile force, so they do not carry a significant amount of load. For this reason, the sample containing the woven fabric showed almost two times less strength than the sample containing the technical embroidery as reinforcement.

The strength characteristics of composites containing roving arranged at an angle of 90° are proportional to the 0° systems. The 4 mm variant was also the most durable, and the UD variant was the smallest. The exception here is the variant containing fabric as a reinforcement. It showed several times greater strength than the other samples. The reason for this is the structure of the fabric itself—half of the threads were placed at an angle of 0° to the tensile force, so they were responsible for the strength of the sample. In other variants, there was no such situation; all rovings were arranged at an angle of 90° to the acting force. Moreover, in the case of these variants, it can be observed that the embroidery improves the strength of the composite. The 8 mm variant had twice the strength of the UD variant, the 2 mm variant—was three times greater, and the 4 mm variant—was two and a half times greater. This is due to the high order of the roving fibers in the sample and the monofilament content in the sample structure.

In the case of arranging the roving at an angle of ±45°, tests were carried out for the variants of 2 mm embroidery, UD and woven fabric. The sample containing woven fabric as reinforcement showed the highest tensile strength. Both reinforcements consisted of four layers, but the woven fabric contained two thread systems in one layer, so the number of load-bearing threads was twice as high. Despite this, the sample with embroidery was only 20% less durable. This indicates that the structure of the embroidery has a positive effect on the strength of the composite. The entire composite is reinforced with a “scaffolding” made of embroidery—the roving is arranged in the plane of the sample, and in a plane perpendicular to it, the fixing monofilament pierces the composite. The UD variant was by far the least resistant to stretching—it showed only one-third of the embroidery strength. Such a low strength was influenced by the chaotic structure of the reinforcement. In the case of embroidery, the roving is situated in a precisely predefined place; therefore, at each point of the sample, the tensile force was acting at an angle of ±45°. On the other hand, the roving in the UD sample was chaotic, so the tensile forces acted at different angles.

Due to the presence of two thread systems perpendicular to each other in the woven fabric, the tensile strength values at the force acting at an angle of 0° and 90° are the same, while the values 0/90° and ±45° are similar to each other. The woven fabric is characterized by greater isotropy of mechanical properties compared to unidirectional arranged embroidery and UD fabrics. Embroidered variants, on the other hand, are characterized by a large range of tensile strength results, depending on the direction of the roving in the sample and, consequently, the direction of the acting force. In the case of embroideries of both 2 and 4 mm, samples containing roving arranged at an angle of 0° to the acting force showed almost six times greater strength than samples containing roving arranged at an angle of 90°. However, when comparing the 8 mm embroidery, the difference was almost ten times. The reason for such a low strength of the 90° systems is the small number of fibers involved in the stretching process. In the case of a force acting at an angle of ±45°, more fibers are involved in the tensile (compared to the force acting at an angle of 90°), and shear forces occur. Therefore, the 2 mm embroidery strength value was almost twice as high for the force acting at an angle of ±45° than for the 90° force. These dependencies allow the most optimal use of the embroidery properties. In the case of woven fabric or UD fabric, the arrangement of fibers in the finished product is often quite random. On the other hand, embroidery allows one to precisely plan and attach the fibers in a specific way. Therefore, if the direction of the forces acting on the product is known, it is possible to optimize its properties by applying embroidery, i.e., arranging the fibers as close as possible to 0° to the force acting on the product. The maximum strain of produced composites is presented in Figure 6.

Considering the composites containing embroidery as reinforcements arranged at an angle of 0°, as in the case of tensile strength, the embroidered system had the highest elongation of 4 mm, then 8 mm, and finally 2 mm. The 4 mm system showed a 25% greater elongation than the 2 mm variant and a 20% greater elongation than the 8 mm variant. The composite with the UD 0° system as reinforcement showed the smallest elongation in this group of reinforcements (0° systems). Its elongation was 30% lower than that of the 4 mm variant. Composite with woven fabric as reinforcement was characterized by the greatest elongation. This is because the roving in the woven fabric structure has a significant crimp, which increases the elongation. The lower elongation of composites containing embroidery as reinforcement is also due to the participation of the fastening thread. In the case of the 0° variants, its maximum elongation was about twice as high as the other samples. This was due to the construction of the fabric itself. It contains two sets of threads perpendicular to each other, so more fibers participate in the stretching process. The strength values of the fabric are the same for both 0° and 90° forces.

Within variants containing 90° embroidered systems, the 4 mm variant also showed the greatest elongation. Elongation values of 2 and 8 mm were almost the same, while the difference was only 1% and statistically insignificant. However, the difference between these variants (2 and 8 mm), and 4 mm, was 15%. 4 mm is, therefore, the optimal stitch length between numerous punctures weakening the structure (2 mm variant) and a reduced proportion of embroidery holding monofilament (8 mm variant). The UD variant in this group (90° systems) showed the lowest elongation among all tested composites. When the composite is stretched perpendicular to the direction of the fibers, the tensile force is transmitted mainly by the resin, which is brittle and has much lower mechanical properties than the fiber. The composite containing woven fabric as reinforcement was approximately five times larger than the other 90° variants. As with the 0° variant, this is due to the greater proportion of fibers in the stretching process.

Comparing all the embroidered and UD patterns, the 2 mm variant with the roving direction at an angle of ±45° showed the greatest elongation. This is because after applying a tensile force, the composite first experiences shear and bending forces. The fibers in the composite first have to move from the ±45° direction to the 0° direction—then they are subjected to a tensile force. For this reason, the fabric variant ±45° showed the greatest elongation among all the composites made.

Considering each direction of the arrangement of the fibers (0°, 90° and ±45°) separately, in each case, it was the UD composite that showed the smallest elongation. In embroidery, the roving is additionally held by a fastening thread, i.e., monofilament. In the case of UD fabric, the roving was attached only to the ends of the sample, and its alignment was not controlled along its entire length. There was the formation of uncontrolled undulations along the length of the sample and the movement of the roving between the layers. The fibers were positioned at different angles to the force applied, which could increase the tensile strength. However, the embroidered systems additionally included monofilament. This element also carried loads, which increased the elongation of embroidered patterns compared to the UD variants.

Within the 0° variants, all three types of embroidery (2, 4 and 8 mm) had similar characteristics of the elongation process (Figure 7); they differed only in maximum strength and elongation. It can be noticed that compared to them, the composite containing woven fabric as reinforcement was characterized by significantly lower tensile strength, greater elongation and lower Young’s modulus. The strength characteristics of the fabric were therefore confirmed by these tests. Even though there were 8 layers of roving in 4 layers of the woven fabric, the embroidery still showed higher tensile strength. However, for this reason, the woven fabric has a greater elongation compared to embroidery. The advantage of embroidery strength is due to the use of monofilament in the embroidery structure, which strengthens the entire structure of the sample.

The appearance of the samples after an attempt to break is illustrated in Figure 8.

In the photos showing the samples after breaking, it can be seen that in the case of composites reinforced with embroidery, the crack was horizontal. Thus, all threads of these systems carried a similar number of loads. The composite containing UD fabric as reinforcement, on the other hand, cracked very unevenly because the roving in this composite was not positioned exactly at an angle of 0° to the acting force.

In the case of the roving arrangement under the direction of 90° in embroidered structures, the embroidery monofilament is the only one to transfer loads of the tensile force—the tensile forces act perpendicular to the direction of flax fibers, and microcracks occur in the resin itself. Therefore, these systems showed the lowest tensile strength of all tested variants (Figure 9). At the same time, with the 90° system, the embroidered samples showed higher tensile strength than the sample containing UD. Due to its structure, the woven fabric showed a much greater elongation and greater tensile strength—a greater proportion of fibers in the stretching process.

The appearance of the samples after an attempt to break is illustrated in Figure 10.

The crack of samples containing embroidery and UD fabric as reinforcement at an angle of 90° to the applied force is a straight line. No threads of the roving are positioned at 0° to the applied force, so there are no visible vertical cracks. In the variants of embroidery, only the monofilament that fixes the embroidery is responsible for the load transfer. On the other hand, in the case of a sample containing woven fabric as a reinforcement, an irregular crack can be observed. Because in this type of sample, two sets of threads, rectangular to each other, participate in the breaking.

In the case of arranging the fibers at an angle of ±45°, the embroidery 2 mm and the woven fabric initially showed very similar strength characteristics (Figure 11). However, the fabric-containing composite eventually achieved more than twice the elongation. The thread fastening the roving in the embroidery did not allow the roving to reach its maximum elongation. The structure of the fabric also contributes to greater elongation of the fabric—the threads have a crimp, so when stretched, they first straighten from the wavy position to a straight one, and then they are stretched. The disordered structure of the UD variant contributed to its low strength parameters, significantly lower than that of other composites.

The appearance of the samples after an attempt to break is illustrated in Figure 12.

In the case of a force acting at an angle of ±45° to the direction of the fibers, cracks can be observed in the direction of the roving. This is particularly evident in the case of a composite containing UD fabric as reinforcement. In this variant, the fibers were arranged most loosely in the structure. In the photo showing the sample containing the embroidery, it can be seen clear remnants of the monofilament forming the embroidery—it was largely responsible for the strength of the sample. Therefore, this variant has proved to be stronger than the variant comprising UD fabric.

The Young modulus of produced composites is presented in Figure 13.

The highest strength and the highest Young’s modulus are achieved in 0° systems composites loaded toward the fibers. This is a characteristic feature of polymer structural composites. Among these systems, the lowest Young’s modulus has the fabric-reinforced composite—it is about two times lower than the other 0° systems. This is a typical property of fabric-reinforced composites, as half of the fibers are at 0° and half at 90° to the force.

The average value of Young’s modulus at the direction of 0° for the UD glass-epoxy composite is about 39 GPa [30]. Therefore, the composites presented in this study containing technical embroidery as reinforcement showed about twice the value of Young’s modulus. The only exception in this variant of reinforcement arrangement was the fabric—due to the arrangement of the fibers. It has Young’s modulus of 30 GPa.

In 90° systems, in glass-epoxy composites, Young’s modulus is about 15 GPa [30]. Therefore, the composites presented in this study also showed a higher Young’s modulus—apart from the UD variant. 4 mm variant—by 70%, 2 mm variant—by 54%, 8 mm variant—37%, fabric reinforced composite—by almost 100%.

For the variants ±45°, Young’s modulus values are also about two times higher than that of standard epoxy-glass composites, for which Young’s modulus is about 10 GPa [30].

During the tests, it was shown that samples containing embroidery as reinforcement, while stretching in the direction of 0°, proved to be more durable than samples containing woven fabric or UD fabric as reinforcement. In the case of the embroidery length of 4 mm, compared to the sample with woven fabric, the strength was about twice as high. In the case of the force acting at an angle of ±45°, the sample containing embroidery as reinforcement showed similar strength values compared to the other variants. Only in the case of a force acting at an angle of 90° samples containing embroidery as reinforcement showed significantly weaker strength values.

### 4.2. Shear Strength Properties/Out-of-Plane Properties

The bending force was applied to the samples at an angle of 90° to the roving direction. The 2 mm variant showed the highest bending strength (Figure 14); then, in turn, the 4 mm, woven fabric, 8 mm and UD material variants. This dependence results from the amount of monofilament reinforcing the sample along its entire length in a plane parallel to the applied bending force. As the stitch length increases, the amount of monofilament in the sample decreases, which translates into lower flexural strength. The composite containing UD material as reinforcement was characterized by the lowest bending strength, almost two and a half times lower than the strongest variant, i.e., 2 mm. This is caused by the chaotic arrangement of fibers in this type of sample—the roving was attached only to the edges of the sample; along its entire length, it underwent undulations and interpenetration between the layers. Each of the embroidery variants produced turned out to be stronger than the woven fabric (8 mm and woven fabric were almost identical in strength), although each of the reinforcement systems (both the embroidery and the woven fabric) consisted of four layers, i.e., in the case of the fabric, it was eight-thread systems. The use of embroidery as reinforcement has a positive effect on the bending strength of the composite. The woven fabric variant was less strong than the 2 and 4 mm variants. This is due to the fact that the thread used to make the zig-zag stitch strengthens the entire composite, contributing to an increase in the interlaminar shear strength.

In this test, the type of crack formed when the bending force is applied is also an important factor. These cracks can be divided into interlaminar shear, flexure (compression and tension) and inelastic deformation. In the case of samples reinforced with UD material (Figure 15) and fabric (Figure 16), the formation of interlaminar shear can be observed. This is the formation of damage typical of composites containing fabric layers as reinforcement. In the case of composites containing embroidery as reinforcement, “flexure, tension” cracks appear (Figure 17). In these cases, the interlaminar shear strength of the system is greater than that at which the fracture was observed. It is faster to crack than to achieve maximum strength. Monofilament is an out-of-plane reinforcement between the layers and prevents the formation of interlayer cracks and delamination of the composite. The types of cracks formed are presented in the photos below. The places where the cracks occur have been marked.

It must be noticed that the short beam test allows knowing the usual S13 property of a laminate, out-of-plane shear strength. It is a very good indication of how the composite laminate should behave out of a plane. Likewise, it is an out-of-plane property testing rather than a usual bending/flexural test; this is the reason for relatively low properties because these properties are related to out-of-plane stress rather than flexural strength.

## 5. Summary

The world turns to the idea of sustainable development. The demand for composites reinforced with natural raw materials is growing, which is why flax roving was used in the research. Technical embroidery contributes to the reduction of the amount of waste generated during the production of composite preforms. In order to confirm the hypothesis that technical embroidery can successfully replace fabrics in the production of composites, strength tests were carried out, and composites containing technical embroidery, UD material and fabric as reinforcement were compared.

The use of embroidery to reinforce composites improves their strength properties compared to unidirectional fiber systems and to woven fabrics.

Based on the conducted research, it can be concluded that vertical stitches prevent the formation of interlayer cracks and that the bending strength of the composite increases with the increase in the stitch density. This fact should be considered while designing finished products containing embroidered patterns as reinforcement. In the case of beam systems, it is recommended to arrange the strengthening medium in the direction of the acting tensile forces. However, in the case of materials with similar longitudinal and transverse dimensions, it is recommended to use lattice-shaped systems. Thus, embroidery can be successfully used to strengthen, e.g., a car body. Composites containing embroidery as reinforcement will also find application in the production of small elements made of expensive materials. Thanks to the Tailored Fiber Placement technology, embroidery significantly reduces production losses, reducing the costs of the entire production.

## 6. Conclusions

As a result of the research, the following conclusions can be drawn:The use of technical embroidery as a composite reinforcement increases its tensile strength in the case of a tensile force acting at an angle of 0° to the sample.The optimal stitch length is 4 mm. Then, the sample has the best tensile strength. Too large (2 mm variant) or too small (8 mm variant) number of needle sticks during embroidery production has a negative impact on the tensile strength of the composite.Considering each direction of reinforcement arrangement, the composite with unidirectional (UD) reinforcement showed lower values of strength and elongation compared to composites reinforced with embroidery. This shows that the use of embroidery affects the arrangement of the fibers in the composite and increases its tensile strength.The use of embroidered reinforcements in the composite, compared to woven fabric reinforcements, increases its bending/out-of-plane strength. As the embroidery density increases, the bending strength of the composite increases.During the bending test, composites containing unidirectional fabric as reinforcement behave similarly to traditional fabric-reinforced composites. The same types of interlayer cracks develop. On the other hand, the use of embroidery is a vertical reinforcement of the layers and prevents the formation of interlayer cracks.The technology of technical embroidery allows optimizing the mechanical values of the composite reinforcement.Composites containing technical embroidery made of flax fibers have a greater Young’s modulus than glass-epoxy composites.

## Figures and Tables

**Figure 1 materials-15-07469-f001:**
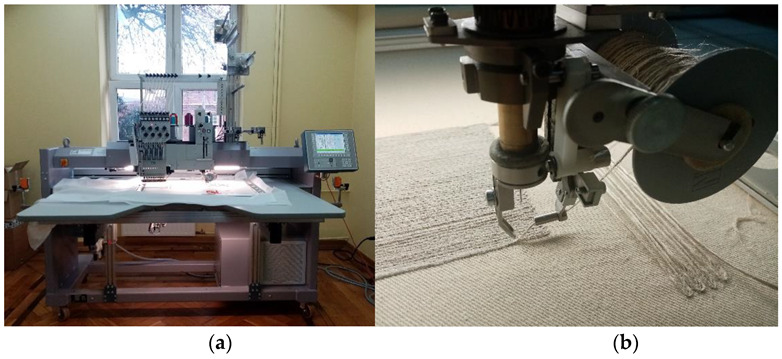
(**a**): ZSK computer embroidery machine; (**b**): W-type head for technical embroidery (own source).

**Figure 2 materials-15-07469-f002:**
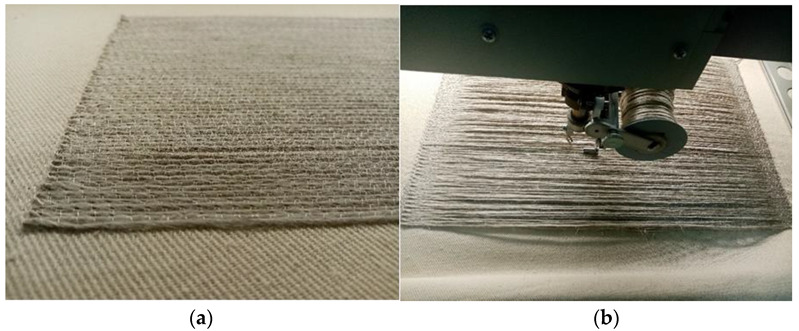
(**a**): technical embroidery; (**b**): UD fabric (own source).

**Figure 3 materials-15-07469-f003:**
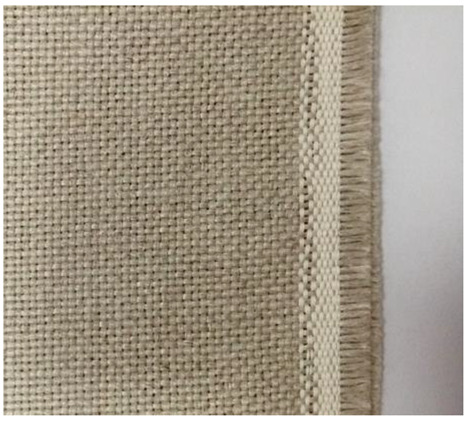
Flax fabric (own source).

**Figure 4 materials-15-07469-f004:**
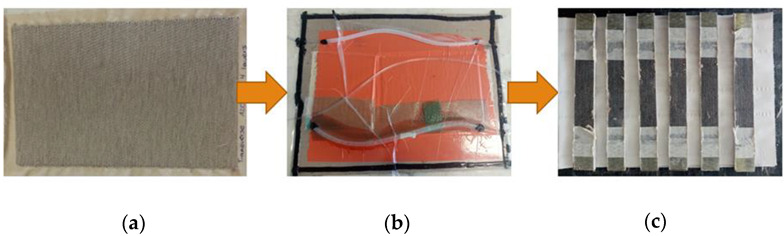
Stages of production of test samples. (**a**): dry sample; (**b**): making a composite; (**c**)**:** cutting into samples of an appropriate size (own source).

**Figure 5 materials-15-07469-f005:**
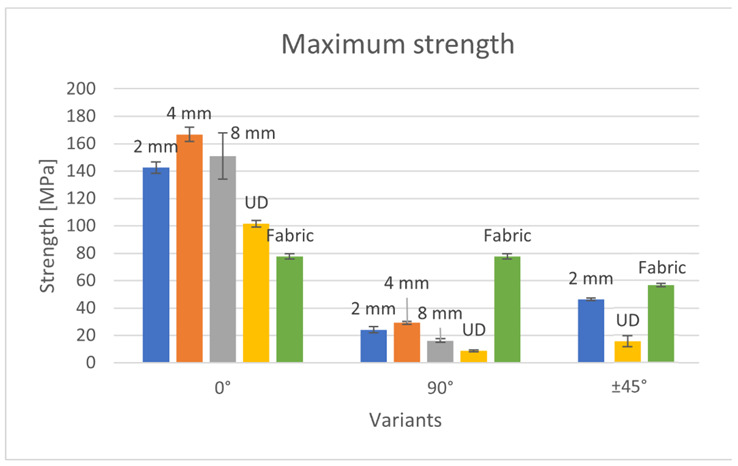
Strength of the produced composites.

**Figure 6 materials-15-07469-f006:**
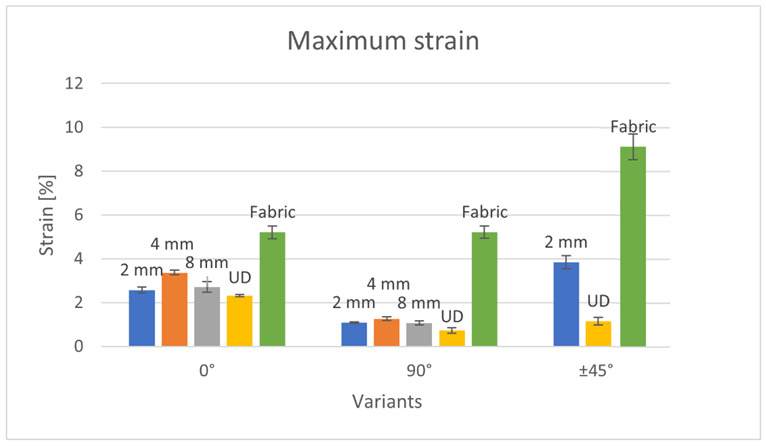
The strain of the produced composites.

**Figure 7 materials-15-07469-f007:**
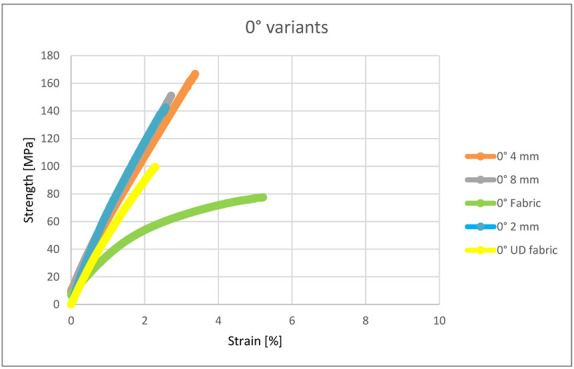
Strength and strain of the 0° variants.

**Figure 8 materials-15-07469-f008:**
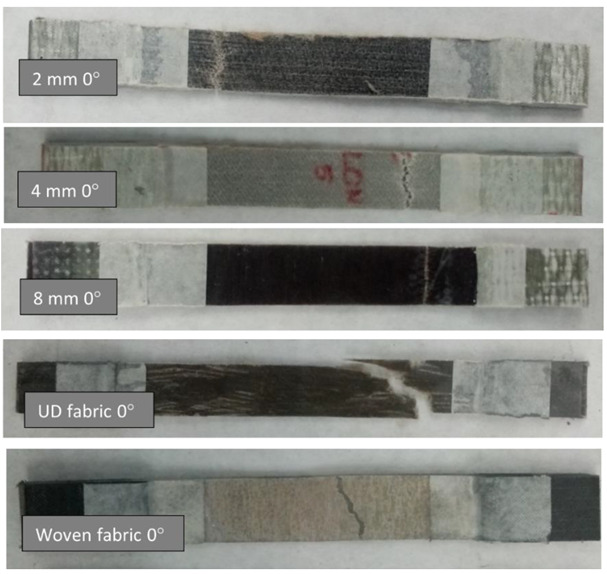
Samples of 0° variant after the break.

**Figure 9 materials-15-07469-f009:**
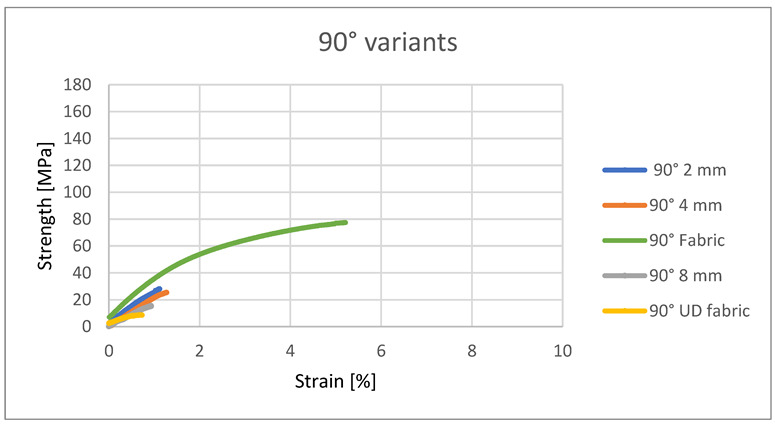
Strength and strain of the 90° variants.

**Figure 10 materials-15-07469-f010:**
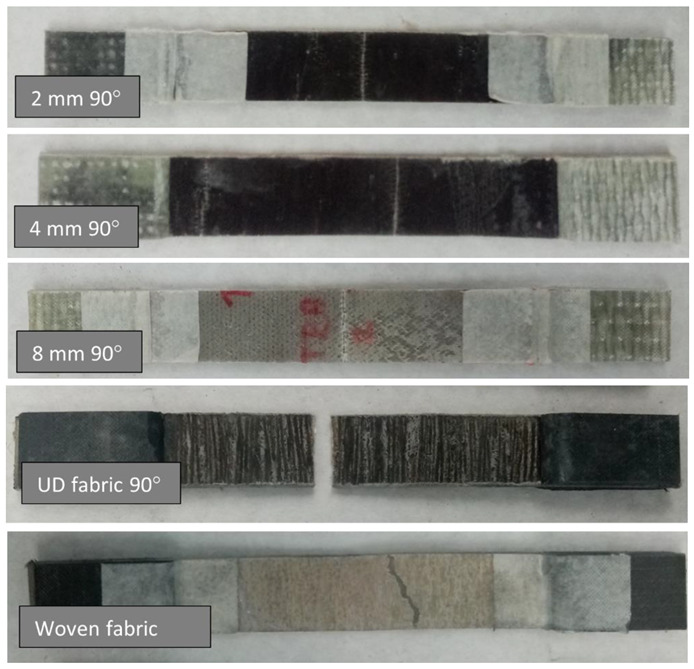
Samples of the 90° variant after the break.

**Figure 11 materials-15-07469-f011:**
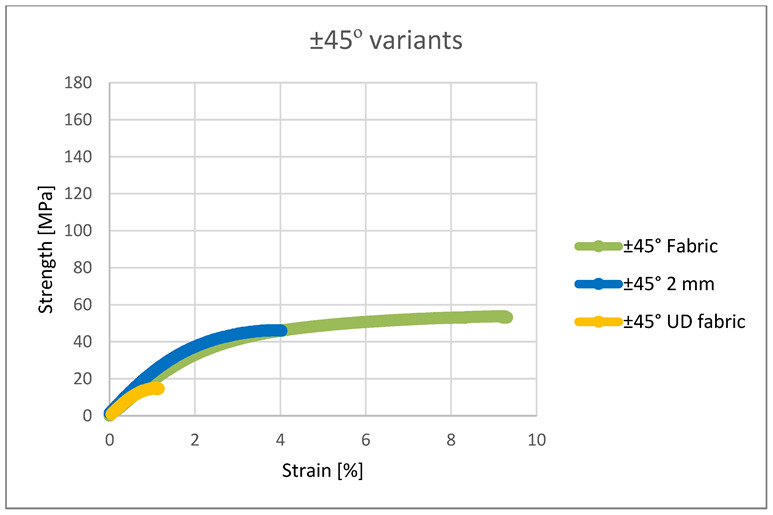
Strength and strain of the ±45° variants.

**Figure 12 materials-15-07469-f012:**
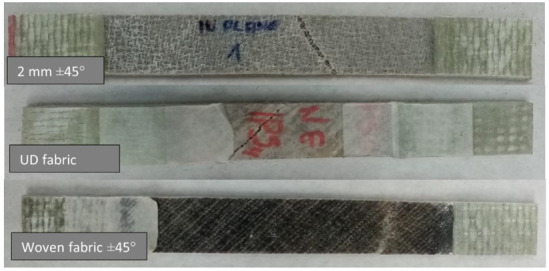
Samples of ±45° variant after the break.

**Figure 13 materials-15-07469-f013:**
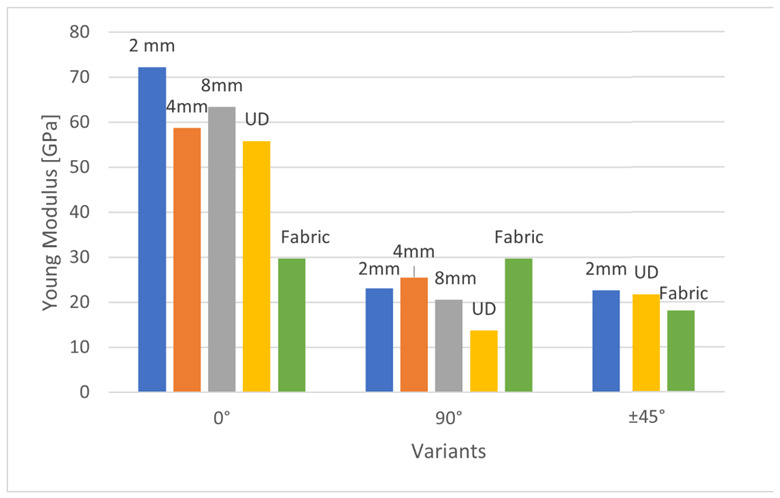
Young modulus of produced composites.

**Figure 14 materials-15-07469-f014:**
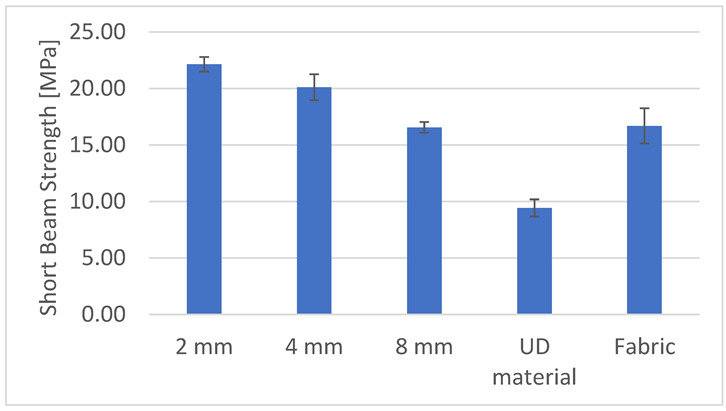
Short beam strength of produced composites.

**Figure 15 materials-15-07469-f015:**
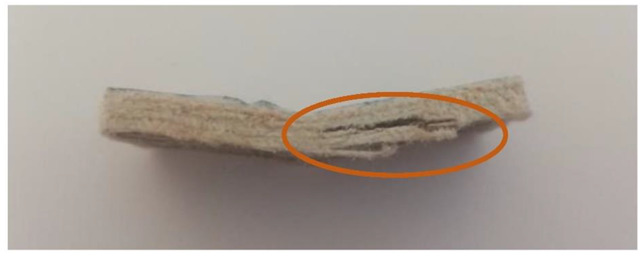
UD fabric sample with an interlaminar shear type of crack (own source).

**Figure 16 materials-15-07469-f016:**
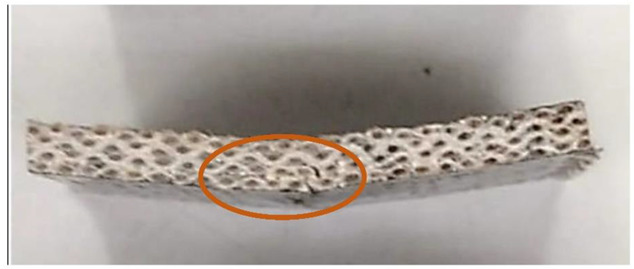
Woven fabric sample with an interlaminar shear type of crack (own source).

**Figure 17 materials-15-07469-f017:**
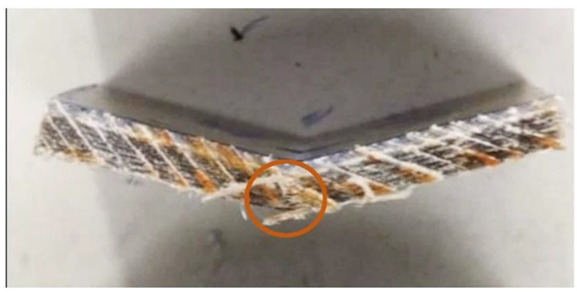
Embroidery 8 mm sample with crack flexure, and tension (own source).

**Table 1 materials-15-07469-t001:** Types of produced reinforcements.

Variant	Name	Orientation of Roving in Each Layer
Embroidery 0° 2 mm	0° 2 mm	││││
Embroidery 0° 4 mm	0° 4 mm	││││
Embroidery 0° 8 mm	0° 8 mm	││││
Embroidery ±45° 2 mm	±45° 2 mm	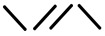
Embroidery 90° 2 mm	90° 2 mm	─ ─ ─ ─
Embroidery 90° 4 mm	90° 4 mm	─ ─ ─ ─
Embroidery 90° 8 mm	90° 8 mm	─ ─ ─ ─
UD fabric 0°	0° UD	││││
UD fabric ±45°	±45° UD	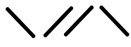
UD fabric 90°	90° UD	─ ─ ─ ─
Fabric 0°/90°	fabric 0°/90°	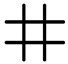
Fabric ±45°	fabric ±45°	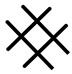

**Table 2 materials-15-07469-t002:** The volume fraction of produced composites.

Variant	Name
Embroidery 0° 2 mm	41%
Embroidery 0° 4 mm	38%
Embroidery 0° 8 mm	34%
Embroidery ±45° 2 mm	40%
Embroidery 90° 2 mm	30%
Embroidery 90° 4 mm	38%
Embroidery 90° 8 mm	34%
UD fabric 0°	56%
UD fabric ±45°	52%
UD fabric 90°	39%
Fabric 0°/90°	42%
Fabric ±45°	44%

**Table 3 materials-15-07469-t003:** Tensile strength tests and tensile elongation test parameters [24].

Parameter	Value
grips distance	100 mm
speed of testing	1 mm/min
sample size	250 mm × 25 mm × 3.5 mm
number of samples	5 of each variant

**Table 4 materials-15-07469-t004:** Bending strength test’s parameters [24].

Parameter	Value
distance between supports	20 mm
speed of testing	1 mm/min
sample size	40 mm × 20 mm × 4 mm
number of samples	5 of each variant

## Data Availability

The data presented in this study are available on request from the corresponding author.

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
