# Peer review of "Comparison of Mechanical Properties of Composites Reinforced with Technical Embroidery, UD and Woven Fabric Made of Flax Fibers"

_materials, 2022, doi:10.3390/ma15217469_

Round 1
Reviewer 1 Report
The authors presented a manuscript pertaining the mechanical properties of composites reinforced fibers
The research presented is relevant to the field of composite material. The motivation was well articulated. The research plan and design of experiment was well described. The authors have explicitly and clearly discussed how their research work contributes to the current field of study. The authors have analyzed prior research related to their work. The authors have explicitly described their methodology and is aligned with currently accepted methodologies. The research showed, the use of technical embroidery as a composite reinforcement increases its tensile strength. Furthermore, the use of embroidery is a vertical reinforcement of the composite and prevents the formation of interlayer cracks. The conclusions was well discussed supported by the results. In terms of organization and clarity, the paper was well constructed. However, a proof-read is required before resubmission.
Below are some questions for the authors:
1) Are implications for future research considered? If appropriate, are implications for practice considered?
2) For the technical embroidery, are there any practical limitations?
3) Any reasons why flax fabric was selected?
4) Figures 5-6 requires a legend for clarity purpose
5) Quality of Figure 6 needs to be improved. The data fonts are too large and it clusters the figure
6) Why in Figure 8-9, the 2 mm variant has a much longer variation of 5 % compared to the others?
7) Figure 10, reduce the cluster of the data in the graph
8) Figure 12 and 13, improve quality of figure to enhance the failure site
9) Why only selected mechanical properties were investigated? Are these tied to some application related?
Reviewer 2 Report
Comments on materials-1960133
The manuscript entitled “COMPARISION OF MECHANICAL PROPERTIES OF COMPOSITES REINFORCED WITH TECHNICAL EMBROIDERY, UD AND WOVEN FABRIC MADE OF FLAX FIBERS” presents the results of strength tests of composites containing technical embroidery, woven fabric and UD fabric as the reinforcement. Each of the samples was made of the same material - flax roving. The samples differed from each other in the arrangement of layers in the reinforcement.
The manuscript is well-written, however, there are several amendments required to be resolved before accepting it for publication which are disclosed below:
· The introduction has some flaws, a more detailed novelty of their work should be clearly addressed which is missing in their Introduction.
· The authors are also suggested to highlight some of their main results/findings in the last paragraph of the Introduction.
· The authors are suggested to write the full forms of all the techniques they used.
· The authors should mention the dimensions of the standard testing samples after cutting them.
· The authors are recommended to discuss more the types of roving orientation they used in each layer and which one is better, which is missing in the main discussion.
· Tables 3 and 4 have a different format style than the other tables. Please check the whole manuscript carefully.
· Some of the Figure’s resolution is very poor. Please improve their resolution. And Figures 6, 7 and 8, 9, 10 are not appropriately adjusted in the manuscript. They are going out of the boundaries. Please check them.
· The authors are highly suggested providing the optical images of all the composite samples after deformation.
· The authors should provide more discussions on the mechanisms for performance strategies, which would be beneficial for readers to understand their significance.
· The authors need to provide a comparison of their work with other published ones to assure the novelty of their work which is lacking in their submitted manuscript.
· The authors are recommended to add a few sentences about the prospective applications of their proposed work before the Conclusions.
· The conclusions are a bit lengthy, please trim them.
· It is also recommended to add some references from recent years of the related work. Some references are important to understand the progress of biomass composite materials and their advantages: Materials Today Communications, 2022, 31, 103858.
· The language expression in the text needs to be carefully checked and revised. There are some grammatical mistakes.
Round 2
Reviewer 2 Report
In the reviewer's pdf version, Figures 5 and 13 are still going out of the boundary, please check them.
The authors have answered all the reviewer's comments and suggestions, it can be accepted for publication.